# Additive Manufacturing and Combustion Characteristics of Polyethylene Oxide/Aluminum/Copper Oxide-Based Energetic Nanocomposites for Enhancing the Propulsion of Small Projectiles

**DOI:** 10.3390/nano13061052

**Published:** 2023-03-15

**Authors:** Ho Sung Kim, Soo Hyung Kim

**Affiliations:** 1Research Center for Energy Convergence Technology, Pusan National University, 2, Busandaehak-ro 63beon-gil, Geumjeong-gu, Busan 46241, Republic of Korea; 2Department of Nano Fusion Technology, College of Nanoscience and Nanotechnology, Pusan National University, 2, Busandaehak-ro 63beon-gil, Geumjeong-gu, Busan 46241, Republic of Korea; 3Department of Nanoenergy Engineering, College of Nanoscience and Nanotechnology, Pusan National University, 2, Busandaehak-ro 63beon-gil, Geumjeong-gu, Busan 46241, Republic of Korea

**Keywords:** nanoenergetic material, additive manufacturing, composite solid propellant, propulsion characteristic, small projectile

## Abstract

The application of nanoscale energetic materials (nEMs) composed of metal and oxidizer nanoparticles (NPs) in thermal engineering systems is limited by their relatively high sensitivity and complex three-dimensional (3D) formability. Polymers can be added to nEMs to lower the sensitivity and improve the formability of 3D structures. In this study, the effect of the addition of polyethylene oxide (PEO; polymer) on the combustion characteristics of aluminum (Al; fuel)/copper oxide (CuO; oxidizer)-based nEMs is investigated. With an increase in the PEO content, the resulting PEO/nEM composites are desensitized to relatively high electrical spark discharges. However, the maximum explosion-induced pressure decreases significantly, and the combustion flame fails to propagate when the PEO content exceeds 15 wt.%. Therefore, the optimal PEO content in a nEM matrix must be accurately determined to achieve a compromise between sensitivity and reactivity. To demonstrate their potential application as composite solid propellants (CSPs), 3D-printed disks composed of PEO/nEM composites were assembled using additive manufacturing. They were cross-stacked with conventional potassium nitrate (KNO_3_)/sucrose (C_12_H_22_O_11_)-based disk-shaped CSPs in a combustion chamber of small rocket motors. Propulsion tests indicated that the specific impulse of KNSU/PEO/nEM (nEMs: 3.4 wt.%)-based CSPs was at a maximum value, which is approximately three times higher than that of KNSU CSPs without nEMs. This suggests that the addition of an optimized amount of polymer to nEMs is beneficial for various CSPs with compromised sensitivity and reactivity and excellent 3D formability, which can significantly enhance the propulsion of small projectiles.

## 1. Introduction

Energetic materials (EMs) are composed of metal fuel and an oxidizer, where internal chemical energy is rapidly converted into thermal energy by a critical external energy input. In particular, nanoscale energetic materials (nEMs) have a higher redox reactivity than microscale EMs owing to their higher specific surface area, shorter diffusion distance, and lower melting point [1,2,3,4,5,6,7,8,9,10,11]. Thus, the heat energy released and explosion pressure of nEMs are expected to be higher when they are ignited. These materials have been applied in various civil and military thermal engineering systems (e.g., explosives, solid propellants, metal welding, and igniters) [12,13,14,15,16].

The propulsion force of projectiles can be changed significantly by varying the shape of the solid propellant grain filled into the combustion chamber [17,18]. Generally, two conventional methods are used for assembling solid propellant grains. First, a template is installed in a combustion chamber filled with solid propellant powder, and then the template is removed after the solid propellant powder has hardened into a bulk grain. Second, solid propellant powders are pressurized in a mold for extrusion with a solid propellant grain [19,20,21]. However, solid propellants can unexpectedly explode due to mechanical impact, friction, and static electricity occurring in the aforementioned template removal and extrusion processes. Therefore, a safer and more versatile method is required for manufacturing solid propellant grains.

Recently, various studies on the additive manufacturing of three-dimensional (3D) structures using polymer/EM-based composites have been conducted [22,23,24,25]. For example, Wang et al. successfully fabricated a 3D structure by spraying a colloidal solution dispersed with polymer/EM-based composites [22]. However, the amount of the colloidal solution was limited by the capacity of the syringe, and the particles in the colloidal solution gradually precipitated over time. Thus, manufacturing a relatively large and uniform 3D structure of polymer/EM-based composites is challenging. Additionally, the solvent used for particle dispersion in the colloidal solution must be removed using a secondary evaporation process. Trevor et al. also reported the successful fabrication of a polymer/EM-based filament using 3D printing technology [23]. However, there was a high risk of explosion because of the overheating of the 3D printing nozzle; friction was also generated between the nozzle and the accumulated materials. Thus, it is difficult to select an EM for the safe 3D printing of solid propellants.

In this study, polymer/nEM-based composites were fabricated using additive manufacturing to safely form 3D-structured solid propellant grains. The physical and combustion characteristics of these polymer/nEM-based composites were systematically investigated in terms of electron microscopic images, explosion-induced pressure, burn rate, and ignition threshold spark energy. Additionally, a series of propulsion tests for small rocket motors, which were charged with potassium nitrate (KNO_3_) and sucrose (C_12_H_22_O_11_) (hereafter KNSU)-based disks and polymer/nEM composite-based disks, were performed to investigate the effect of the 3D-structured polymer/nEM composites on the resulting propulsion force of small projectiles.

## 2. Materials and Methods

### 2.1. Additive Manufacturing of PEO/nEM Composite-Based 3D Structures

In this study, a polyethylene oxide (PEO)/nEM composite-based colloidal solution was prepared by dispersing aluminum (Al) and copper oxide (CuO) nanoparticles (NPs) in a PEO/ethanol solution, as shown in Figure 1. Commercially available Al (NT base Inc., Yongin Si, South Korea) and CuO (NT base Inc., Yongin Si, South Korea) NPs with average diameters of 103 ± 18 and 138 ± 19 nm were used without further treatment. To prepare an nEM-dispersed colloidal solution, Al and CuO NPs were mixed with an Al:CuO ratio of 3:7 in wt.% (i.e., fuel-to-oxidizer ratio, Φ = 1.78), and then the colloidal solution was sonicated for 1 h at 200 W and 40 kHz. Additionally, PEO (Sigma-Aldrich, St. Louis, MO, USA) dispersed in ethanol was sonicated at 200 W and 40 kHz at 50 °C until the PEO was completely dissolved. The nEM-dispersed colloidal and PEO solutions were then mixed and stirred at 600 RPM for 1 h to prepare the PEO/nEM composite-dispersed colloidal solution for additive manufacturing. An additive manufacturing system was developed using various parts, including a colloid vessel, 3D-printer axis, syringe needle, and a heater. Specifically, a syringe needle with an inner diameter of 1.2 mm and a needle adapter were installed at the tip of a liquid-transporting tube, which was connected to a vessel containing the PEO/nEM colloidal solution. The vessel was placed on a magnetic stirrer to mix the PEO/nEM solution continuously. The amount of the PEO/nEM colloidal solution transported to the needle was adjusted by varying the nitrogen gas pressure supplied to the vessel. The PEO/nEM solution dropped from the needle was dried rapidly on the surface of a nitrocellulose (NC)-coated glass stage, and then it was finally formed into accumulated 3D structures (i.e., PEO/nEM disks), which were lifted off by immersing them in an acetone solution to remove the NC-based sacrificial layer coated on the glass. In addition, the potassium nitrate and sucrose (hereafter named as KNSU)-based composite solid propellants (CSPs) were made into 3D disks using a mold and mount compressor operated under a pressure of ~300 MPa. The as-prepared 3D disks composed of KNSU and PEO/nEMs, respectively, were cross-stacked in the combustion chamber of a small rocket motor for propulsion tests (see Figure 1).

### 2.2. Measurement of Physical and Combustion Characteristics of PEO/nEM Composites

The physical and combustion characteristics of the as-prepared PEO/nEM composites were evaluated using various methods. The microstructures and elemental distributions of the PEO/nEM composites were analyzed using a field-emission scanning electron microscope (FE-SEM; acceleration voltage = 10 kV, S-47, Hitachi, Tokyo, Japan), and the cross-section and density of the 3D-printed PEO/nEM structures were analyzed using an optical microscope (BX60MF5, Olympus, Tokyo, Japan). The explosion-induced pressure of the PEO/nEM composites was determined using a pressure-cell tester (PCT), in which the PEO/nEM composites were manually ignited with a hotwire, and a pressure sensor (113A03, PCB Piezotronics, Depew, NY, USA) was installed to determine the explosion pressure. An electrostatic discharge simulator (ESD; capacitance = 330 pF, accelerating voltage = 30 kV, maximum discharge energy = 150 mJ, KES4021, Kikuchi Electronics Corp., Tokyo, Japan) test was performed to determine the electrical discharge sensitivity of the PEO/nEM composites. The burn rates of the PEO/nEM composites were measured using a high-speed camera (frame rate = 30,000 FPS, FASTCAM SA3 120 K, Photron Co. Ltd., Tokyo, Japan). The propulsion characteristics of a small rocket motor charged with 3D-printed PEO/nEM composite disks were determined using a load cell (PW2D; HBM, Darmstadt, Germany).

## 3. Results and Discussion

Figure 2 shows the FE-SEM images of the Al NPs (fuel), CuO NPs (oxidizer), and PEO/nEM (i.e., Al+CuO NPs) composites. As shown in Figure 2a,b, Al and CuO NPs are spherical structures with an average diameter of ~103 ± 18 nm and 138 ± 19 nm, respectively. However, the Al and CuO NPs were held together and formed microsized clusters in the PEO/nEM composites because of the chain bonding of PEO (see Figure 2c).

Energy-dispersive X-ray spectroscopy (EDS) analysis of the 3D-printed PEO/nEM composite structure was performed to examine the distribution of elemental compositions. Figure 3a,b show the PEO/nEM composite-based 3D-printed rod (width: 3 mm, height: 3 mm, length: 70 mm). Figure 3c shows the FE-SEM image of the cross-sectional area of the PEO/nEM (PEO:nEMs = 10:90 wt.%) composite-based 3D-printed rod with densely accumulated layers. As the results of EDS analysis, the elemental mapping images (Figure 3d–f) show that the distribution of C, Al, and Cu elements were relatively uniform in the PEO/nEM composite matrix. This suggests that the 3D-printed energetic composite structures (ECSs) fabricated in this study have relatively uniform distributions of elemental compositions in the 3D composite matrix.

The number of printing layers was increased to examine the growth of 3D-printed ECSs (i.e., PEO:nEMs = 10:90 wt.%). From the results of optical microscope observation (see Figure 4a–e), the resulting thickness of 3D-printed ECSs increased significantly with the number of printing layers. Additionally, numerous small pores in the cross-sectional area and on the surfaces of the 3D-printed ECSs were observed. The calculated absolute and theoretical densities of 3D-printed ECSs are 2.2–2.6 g/cm^3^ and 65–73% TD, respectively, with 20 to 100 printing layers, as shown in Figure 4f. This suggests that the Al and CuO NPs were densely distributed and bound by PEO in the 3D-printed ECSs.

As shown in Figure 5, PCT measurement was performed to analyze the explosion-induced pressure rise as a function of elapsed time. The maximum pressure caused by the composite explosion decreased significantly with increasing PEO content in the PEO/nEM matrix. The rise time to reach the maximum pressure increased significantly with PEO content. This suggests that the addition of PEO to the nEM composites suppresses the resulting redox and aluminothermic reactions because the increased amount of PEO disturbs oxygen transfer between the Al and CuO NPs to some extent.

To examine the electrical discharge sensitivity of the PEO/nEM composites, an electrical spark was repeatedly loaded into the composites with a maximum electrical energy of ~150 mJ (i.e., applied voltage: ~30 kV, stored electrical power: ~330 pF), which exceeds the spark energy generally generated by human activity (~8.3 mJ), as shown in Figure 6. An electrical discharge tip was placed on top of the PEO/nEM composites (loading mass: ~5 mg) at a distance of ~1 mm. An electrical ignition test was performed by varying the acceleration voltage. When nEMs without PEO addition (PEO:nEMs = 0:100 wt.%) were tested, the ignition threshold spark energy was determined to be ~2.4 mJ. However, when ~5 wt.% PEO content was added, the resulting ignition threshold spark energy for PEO/nEM composites (i.e., PEO:nEMs = 5:95 wt.%) increased significantly up to ~140 mJ. Ignition was not observed at the maximum spark energy of ~150 mJ for the PEO/nEM composites when the PEO content was >10 wt.%. This suggests that the addition of PEO can reduce the electrical discharge sensitivity of nEMs significantly, thus improving the handling stability of PEO/nEM composites.

As representative 3D ECSs, serpentine patterns incorporated with the PEO/nEM composites were printed on a glass substrate, as shown in Figure 7a. The serpentine patterns had a width of ~2.4 mm, height of ~1.0 mm, and total length of ~260 mm. Highly porous surfaces were observed with an optical microscope. The outer end of the serpentine pattern was initially ignited using a hotwire to examine the burn rate of the PEO/nEM composite-based serpentine pattern. The combustion flame continuously propagated along the serpentine pattern to the other end without stopping for the PEO/nEM composites incorporated with <10 wt.% PEO content. The burn rate of the serpentine pattern was calculated by dividing the total length by the total time taken for the flame to travel from one end to the other. The burn rate of the PEO/nEM composite-based 3D pattern decreased with increasing PEO content. When the PEO content added was more than 15 wt.%, the combustion flame initially caused using a hotwire was extinguished in the middle of the 3D pattern. This confirms that the presence of a large amount of PEO in the PEO/nEM matrix can interrupt the resulting redox and aluminothermic reactions of the nEMs (i.e., Al and CuO NPs).

The propulsion characteristics of small rocket motors (see Figure 8a,b) charged with conventional KNSU (i.e., KNO_3_+C_12_H_22_O_11_)-based composite solid propellants (CSPs) and PEO/nEM composites (see Figure 8c) were investigated. The KNSU-based CSPs were made into a disk shape with an inner diameter of ~3 mm, outer diameter of ~10 mm, and thickness of ~2 mm using a mold and mount compressor operated under a pressure of ~300 MPa. Furthermore, the PEO/nEM composites were made into a disk shape with an inner diameter of ~3 mm, outer diameter of ~10 mm, and thickness of ~0.3 mm using a 3D printer (see the inset of Figure 8c). To examine the effect of the amount of nEMs on the propulsion of KNSU-based CSPs, PEO/nEM composite-based 3D-printed disks were inserted between the KNSU-based disks, as shown in Figure 8c.

A series of propulsion tests for small rocket motors were performed with the as-prepared cross-stacked KNSU- and PEO/nEM-based 3D disks, which were charged into the combustion chamber of the small rocket motors. After fused ignition was initiated at the exit of the nozzle, the subsequent combustion processes were recorded using a video camera. Figure 9 shows the resulting images of the propulsion tests for KNSU-based CSPs incorporated with different amounts of PEO/nEM composites (see Figure 8c). As the total amount of nEM composite increased in the KNSU-based CSPs, more combustion gases were generated. However, the total combustion time decreased significantly with the increase in the amount of nEMs in the KNSU-based CSPs. This implies that the addition of more PEO/nEM composites accelerates the combustion processes of KNSU-based CSPs.

To observe the propulsion characteristics of the small rocket motor charged with KNSU/PEO/nEM-based 3D CSPs, the thrust trace over time was measured using a load cell installed in the propulsion tests. Herein, the specific impulse is a representative index for the performance evaluation of propellants and refers to the thrust generated when 1 kg of propellant is burned for 1 s. The total impulse (i.e., the total area of the thrust trace over time) was divided by the gravitational acceleration and weight of CSPs. As the total amount of nEMs increased in the KNSU/PEO/nEM-based 3D CSPs, the maximum thrust increased significantly. However, the thrust duration decreased considerably, as shown in Figure 10a. The resulting specific impulse of the KNSU/PEO/nEM-based 3D CSPs reached maximum values at 3.4 wt.% of nEMs. However, it suddenly decreased when excessive nEM of >4.7 wt.% was added to the KNSU/PEO/nEM-based 3D CSPs (see Figure 10b). This is because the aluminothermic reaction of excess nEM enhances the combustion processes of KNSU-based CSPs significantly, which is rapidly consumed in such a short time. This suggests that the addition of an optimized amount of nEM composites into conventional KNSU-based CSPs can enhance the propulsion characteristics significantly by promoting the combustion processes in small rocket motors.

## 4. Conclusions

In this study, 3D PEO/nEM composite structures were successfully fabricated using additive manufacturing, which enabled the uniform distribution of Al NPs (fuel), CuO NPs (oxidizer), and PEO (polymer binder). First, when the PEO content was ≤5 wt.%, the PEO/nEM composite 3D structures were easily broken due to the weak chain bonding of the PEO polymer binder. As the PEO content increased (i.e., PEO content > 5 wt.%), the electrical discharge sensitivity of the nEM composites reduced significantly, implying that the handling stability of the as-prepared PEO/nEM composites improved. However, the maximum explosion-induced pressure and burn rate of the nEM composites decreased significantly with increasing PEO content, which presumably perturbed the redox and aluminothermic reactions of the nEM composites. The combustion flame that was observed on the ignited PEO/nEM composites did not continuously propagate when excessive PEO (≥15 wt.%) was added to the nEM composites. This suggests that the optimal PEO content (i.e., PEO:nEMs = 10:90 w.t%) should be determined to achieve a compromise with the formability, sensitivity, and explosive reactivity of PEO/nEM-based composites and 3D structures. Second, to examine the effect of nEMs on the propulsion characteristics of KNSU-based 3D CSPs, the as-prepared PEO/nEM-based 3D-printed disks were cross-stacked with the KNSU-based disks in the combustion chamber of small rocket motors. After a series of propulsion tests, the specific impulse of the KNSU/PEO/nEM-based 3D CSPs was observed to increase with nEM content up to ~3.4 wt.% in the KNSU-based CSPs. However, it decreased significantly when the nEM content exceeded ~4.7 wt.% in the KNSU-based CSPs. This suggests that the presence of a critical amount of nEMs can promote the combustion of KNSU-based 3D CSPs, thereby improving the propulsion of small rocket motors. However, excessive nEMs can deteriorate the specific impulse by rapidly consuming all the KNSU-based CSPs in such a short time owing to the extremely high heat energy generated by the active redox and aluminothermic reactions of excessive nEMs.

## Figures and Tables

**Figure 1 nanomaterials-13-01052-f001:**
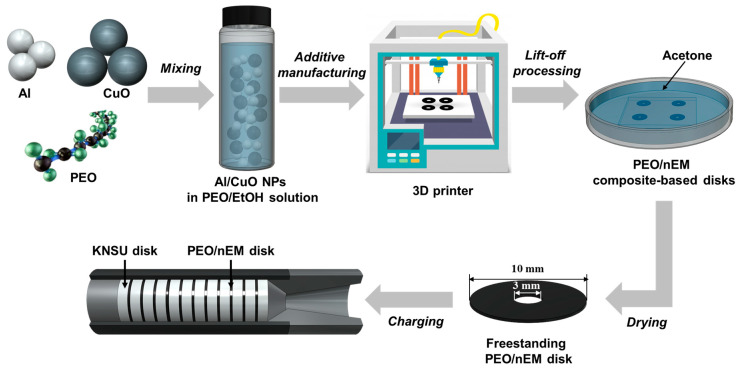
Schematic of the fabrication of PEO/nEM composite-based 3D disks and their charging into the combustion chamber of a small rocket motor.

**Figure 2 nanomaterials-13-01052-f002:**
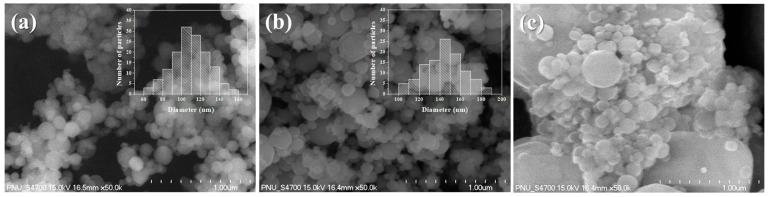
FE-SEM images of (**a**) Al NPs, (**b**) CuO NPs, and (**c**) PEO/nEM composites (PEO:nEMs = 10:90 wt.%).

**Figure 3 nanomaterials-13-01052-f003:**
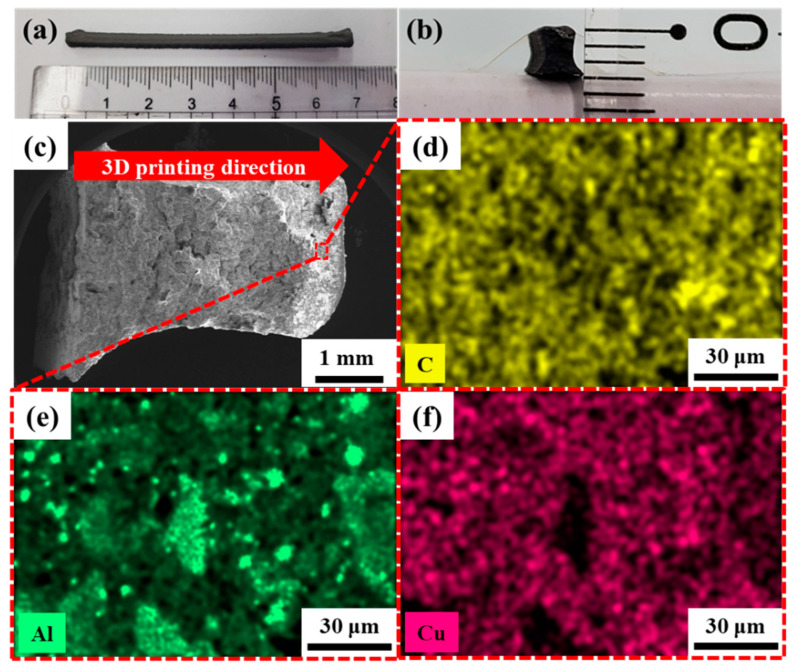
Photographs of the (**a**) top and (**b**) cross-sectional views of PEO/nEM (PEO:nEMs = 10:90 wt.%) composite-based 3D-printed rod structure. (**c**) Low-resolution SEM image of cross-sectional area of PEO/nEM composite-based 3D-printed rod, and the elemental mapping images of (**d**) carbon (C), (**e**) aluminum (Al), and (**f**) copper (Cu) elements, respectively, at a specific area of the cross-section of the PEO/nEM composite-based 3D-printed rod.

**Figure 4 nanomaterials-13-01052-f004:**
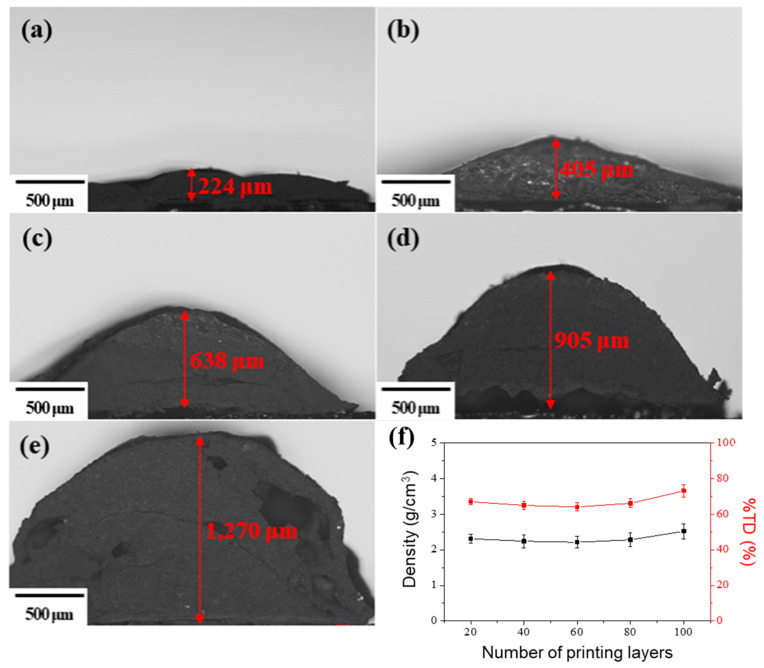
Optical microscope images of cross-section of 3D-printed ECSs (i.e., PEO:nEMs = 10:90 wt.%) with the number of printing layers (**a**) 20, (**b**) 40, (**c**) 60, (**d**) 80, and (**e**) 100, and (**f**) graphs of calculated and theoretical densities of 3D-printed ECSs as a function of the number of printing layers.

**Figure 5 nanomaterials-13-01052-f005:**
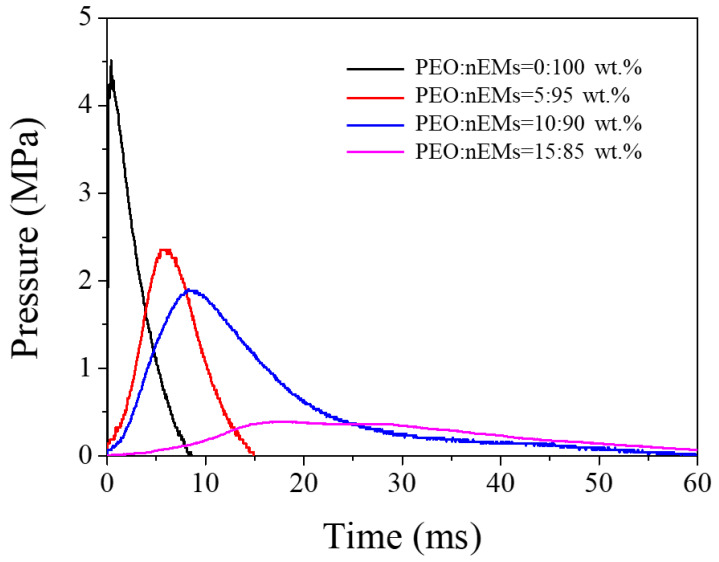
Explosion-induced pressure traces of ignited PEO/nEM composites with different PEO contents in the PCT system.

**Figure 6 nanomaterials-13-01052-f006:**
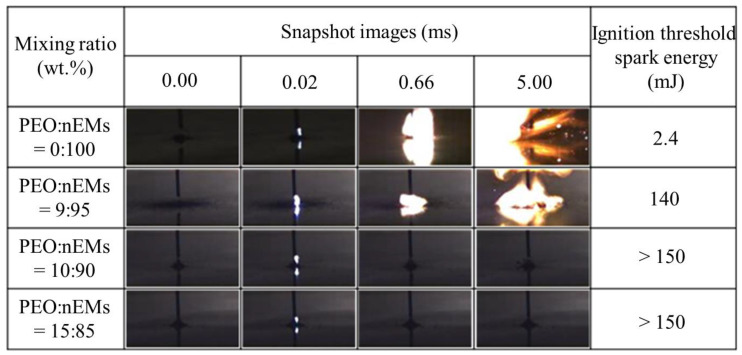
A series of still images of electrical spark ignition tests for PEO/nEM composites incorporated with different PEO contents.

**Figure 7 nanomaterials-13-01052-f007:**
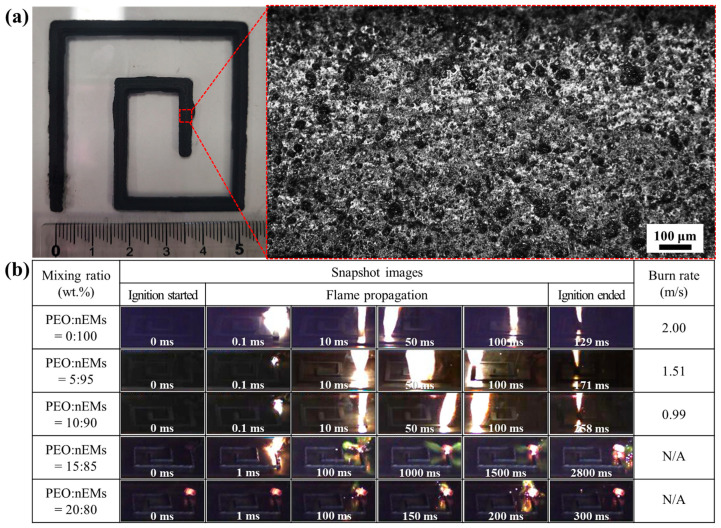
(**a**) Photograph and optical microscope images of 3D-printed serpentine pattern incorporated with PEO/nEM composites. (**b**) Images of the ignition and combustion flame propagation of PEO/nEM composite-based serpentine patterns. (The burn rates of PEO/nEM composites incorporated with >15 wt.% of PEO contents are not available due to the stopping of continuous flame propagation.).

**Figure 8 nanomaterials-13-01052-f008:**
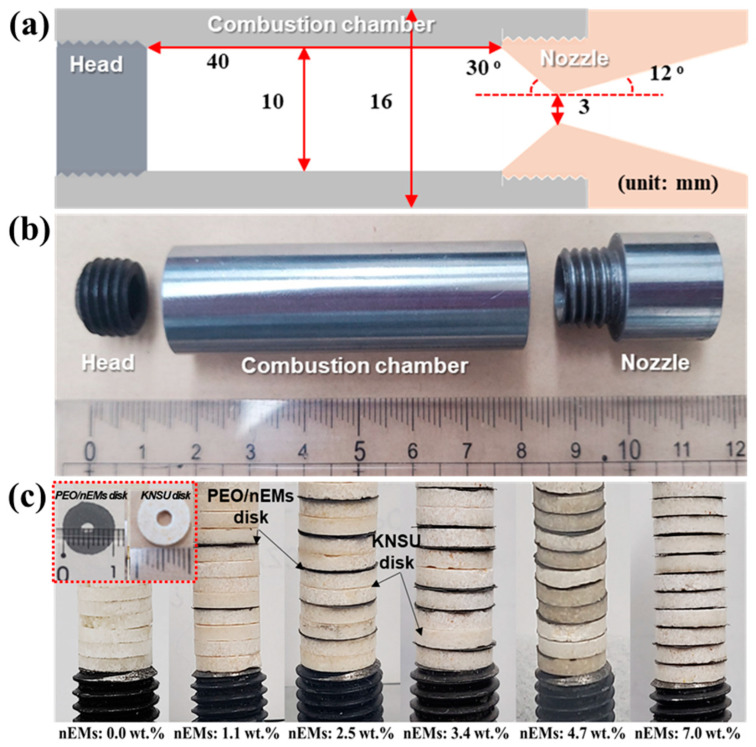
(**a**) Schematic and (**b**) photograph of small rocket motor. (**c**) Photographs of stacked KNSU-based disks inserted with different amounts of PEO/nEM-based 3D-printed disks. (Note: The pure amount of nEMs added in the KNSU/PEO/nEM-based CSPs is presented under each photograph.).

**Figure 9 nanomaterials-13-01052-f009:**
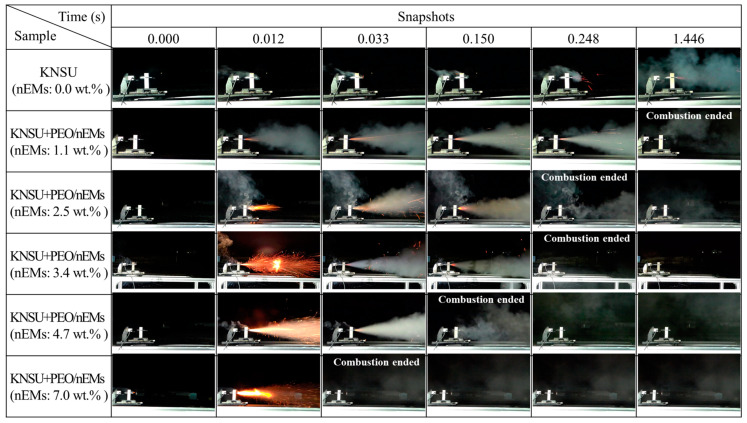
A series of still images of propulsion tests for small rocket motors charged with KNSU-based disks inserted with different amounts of PEO/nEM composite-based 3D-printed disks.

**Figure 10 nanomaterials-13-01052-f010:**
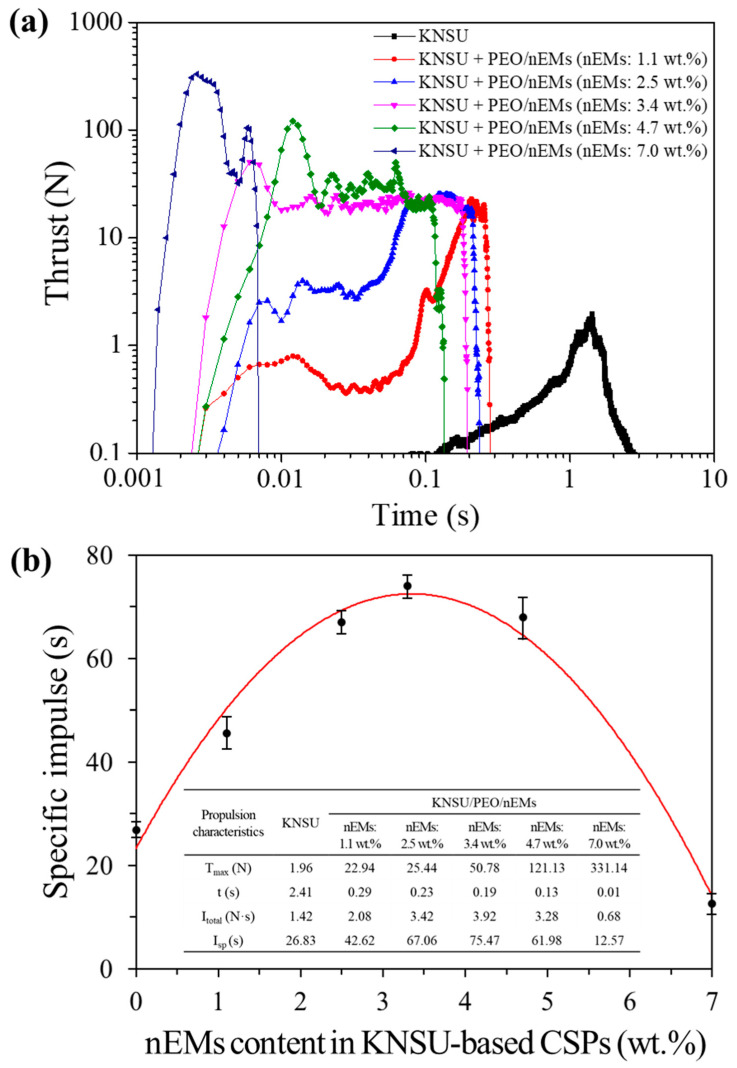
(**a**) Thrust traces and (**b**) specific impulses of KNSU/PEO/nEM-based 3D CSPs incorporated with various nEM contents (Inset shows the specific values of propulsion characteristics of KNSU/PEO/nEM-based 3D CSPs, where T_max_: maximum thrust, t: thrust duration, I_total_: total impulse, and I_sp_: specific impulse).

## Data Availability

The data supporting the findings of this study are available upon reasonable request from the authors.

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
