# Peer review of "Additive Manufacturing and Combustion Characteristics of Polyethylene Oxide/Aluminum/Copper Oxide-Based Energetic Nanocomposites for Enhancing the Propulsion of Small Projectiles"

_nanomaterials, 2023, doi:10.3390/nano13061052_

Round 1

Reviewer 1 Report

Dear Authors,

in your interesting manuscript, the following points should be added/changed to further improve it:

- lines 87-88: Averages +- standard deviations must be given in brackets or with doubled units, e.g. (103 +- 18) nm, else the average wouldn't have a unit. Averages have the same accuracy as their standard deviations; the latter have max. 2 digits (ditto in line 133 etc.)

- line 147, "very uniform distributions of elemental compositions": In Fig. 3 there are some Al-rich areas with low amount of Cu visible, so maybe you should state this slightly differently.

- Fig. 3 (d)-(f): The scale bars and their descriptions are very hard to read.

- Fig. 4: I am wondering about the shapes of these cross-sections. Why are the samples not cylinders with an even top layer? And are there large air cavities vibible in Figs. 4d and 4e? How did you get flat disks from them, as shown in Fig. 1 (cf. lines 226-227)? How many specimens were printed per number of printing layers, and how does the sample hight change with the number of layers?

- Fig. 5: Are the integrals over these curves always identical, or is one (the blue one?) highest?

- Please have a native speaker correct the text and format the references according to the journal template.

Reviewer 2 Report

An interesting work of Ho Sung Kim and Soo Hyung Kim is  devoted to the development of an approach for simple and safe 3D molding of nanoscale energetic materials (nEMs). For this purpose, the Authors proposed adding polyethylene oxide (PEO) to nEMs (Al +CuO) followed by 3D printing from a colloidal solution. The Authors carried out a comprehensive study of the PEO/nEMs composites obtained by them, including determining the effect of the peo content in the composite on such parameters as burning rate, electric discharge sensitivity, etc. They also demonstrated the use of 3D molded PEO/nEMs to create a sandwich type rocket propellant based on potassium nitrate/sucrose composite, for which the optimal content of nEMs was determined.

I recommend this work for publication after the following small comments have been addressed:

1) for the PEO/nEMs composite disscused in figs 2c, 3, and 4 the PEO content should be mentioned, as well as the PEO content should be specified for PEO/nEMs composite used with KNSU-based CSPs 

2) It has been shown that a higher PEO content in PEO/nEMs leads to slower burning rate. On the other hand, the nEMs content over 5 wt% in KNSU+PEO/nEMs CSPs leads to the too fast combustion of the CSP and as a result to a lower specific impuls. It is interesting whether it is possible to control the burning rate of KNSU+PEO/nEMs CSP by changing the PEO content? How does it affect the thrust?

3) how does the inhomogeneity of PEO distribution over KNSU CSP affect the combustion efficiency? uniformity of distribution increases with an increase in the mass fraction of nEMs, which leads to a significant acceleration of fuel combustion. Does a more uniform distribution of nEMs contribute to faster combustion of the fuel, or is it just a matter of mass fraction of nEMs? Is it possible to print PEO/nEMs directly on KNSU discs? At first glance, this simplifies the control of the mass fraction of the nEMs and makes the distribution more uniform even at a low nEMs fraction of about 1-2%

4) I recommend to rewrite the conclusins in order to make them more specific, so that they more clearly reflect the most optimal parameters for the system under study 

Reviewer 3 Report

The manuscript entitled “nanomaterials-2260776” dealing with additive manufacturing has been reviewed. The paper has been nicely written but needs significant improvement. Please follow my comments.

1.     What is the main research question for this research work?

2.     Please specify the additive manufacturing method that you used in your paper. I assume it is material extrusion.

3.     What is the future direction of this work?

4.     Please explain “Figure 1. Schematic of the fabrication of PEO/nEMs comp…” Currently, it is not clear.

5.     Please update the introduction with the new publications in the field. Authors are encouraged to read and add the following two new papers in the field.

·       Material extrusion additive manufacturing of 17–4 PH stainless steel: effect of process parameters on mechanical properties

·       Review of quality issues and mitigation strategies for metal powder bed fusion

6.     Additive manufacturing has many advantages over the conventional manufacturing method which can be highlighted in your paper. Please read the following manuscript and add it to the literature to show how additive manufacturing is comparable with conventional manufacturing.

·       Laser subtractive and laser powder bed fusion of metals: review of process and production features

7.     Please proofread the paper.
